# Approximate Bayesian Computation for Discrete Spaces

**DOI:** 10.3390/e23030312

**Published:** 2021-03-06

**Authors:** Ilze A. Auzina, Jakub M. Tomczak

**Affiliations:** Department of Computer Science, Faculty of Science, Vrije Universiteit Amsterdam, De Boelelaan 1111, 1081 HV Amsterdam, The Netherlands; ilze.amanda.auzina@gmail.com

**Keywords:** Approximate Bayesian Computation, differential evolution, MCMC, Markov kernels, discrete state space

## Abstract

Many real-life processes are black-box problems, i.e., the internal workings are inaccessible or a closed-form mathematical expression of the likelihood function cannot be defined. For continuous random variables, likelihood-free inference problems can be solved via Approximate Bayesian Computation (ABC). However, an optimal alternative for discrete random variables is yet to be formulated. Here, we aim to fill this research gap. We propose an adjusted population-based MCMC ABC method by re-defining the standard ABC parameters to discrete ones and by introducing a novel Markov kernel that is inspired by differential evolution. We first assess the proposed Markov kernel on a likelihood-based inference problem, namely discovering the underlying diseases based on a QMR-DTnetwork and, subsequently, the entire method on three likelihood-free inference problems: (i) the QMR-DT network with the unknown likelihood function, (ii) the learning binary neural network, and (iii) neural architecture search. The obtained results indicate the high potential of the proposed framework and the superiority of the new Markov kernel.

## 1. Introduction

In various scientific domains, an accurate simulation model can be designed, yet formulating the corresponding likelihood function remains a challenge. In other words, there is a simulator of a process available that, when provided an input, returns an output, but the inner workings of the process are not analytically available [1,2,3,4,5]. Thus far, the existing tools for solving such problems are typically limited to continuous random variables. Consequently, many discrete problems are reparameterized to continuous ones via, for example, the Gumbel-softmax trick [6] rather than being solved directly. In this paper, we aim at providing a solution to this problem by translating the existing likelihood-free inference methods to discrete space applications.

Commonly, likelihood-free inference problems for continuous data are solved via a group of methods known under the term Approximate Bayesian Computation (ABC) [2,7]. The main idea behind ABC methods is to model the posterior distribution by approximating the likelihood as a fraction of accepted simulated data points from the simulator model, by the use of a distance measure δ and a tolerance value ϵ. The first approach, known as the ABC-rejection scheme, has been successfully applied in biology [8,9], and since, then many alternative versions of the algorithm have been introduced, with the three main groups represented by Markov Chain Monte Carlo (MCMC) ABC [10], Sequential Monte Carlo (SMC) ABC [11], and neural network-based ABC [12,13]. In the current paper, we focus on the MCMC-ABC version [14] for discrete data application, as it can be more readily implemented and the computational costs are lower [15]. Thus, the efficiency of our newly proposed likelihood-free inference method will depend on two parts, namely (i) on the design of the proposal distribution for the MCMC algorithm and (ii) the selected hyperparameter values for the ABC algorithm.

Our main focus is on optimal proposal distribution design as there is no “natural” notion of the search direction and scale for discrete data spaces. Hence, the presented solution is inspired by Differential Evolution (DE) [16], which has been shown to be an effective optimization technique for many likelihood-free (or black-box) problems [17,18]. We propose to define a probabilistic DE kernel for discrete random variables that allows us to traverse the search space without specifying any external parameters. We evaluate our approach on four test-beds: (i) we verify our proposal on a benchmark problem of the QMR-DTnetwork presented by [19]; (ii) we modify the first problem and formulate it as a likelihood-free inference problem; (iii) we assess the applicability of our method for high-dimensional data, namely training binary neural networks on MNIST data; (iv) we apply the proposed approach to Neural Architecture Search (NAS) using the benchmark dataset proposed by [20].

The contribution of the present paper is as follows. First, we introduce an alternative version of the MCMC-ABC algorithm, namely a population-based MCMC-ABC method, that is applicable to likelihood-free inference tasks with discrete random variables. Second, we propose a novel Markov kernel for likelihood-based inference methods in a discrete state space. Third, we present the utility of the proposed approach on three binary problems.

## 2. Likelihood-Free Inference and ABC

Let x∈X be a vector of parameters or decision variables, where $X=R^{D}$ or $X={\left\{0,1\right\}}^{D}$, and $y\in R^{M}$ is a vector of observable variables. Typically, for a given collection of observations of *y*, $y_{data}={\left\{y_{n}\right\}}_{n=1}^{N}$, we are interested in solving the following optimization problem (we note that the logarithm does not change the optimization problem, but it is typically used in practice):(1)$$x^{*}=\arg\max\ln p\left(y_{data}|x\right),$$
where $p(y_{data}|x)$ is the likelihood function. Sometimes, it is more advantageous to calculate the posterior:(2)$$\ln p\left(x|y_{data}\right)=\ln p\left(y_{data}|x\right)+\ln p\left(x\right)-\ln p\left(y_{data}\right),$$
where p(x) denotes the prior over *x* and $p(y_{data})$ is the marginal likelihood. The posterior $p(x|y_{data})$ could be further used in Bayesian inference.

In many practical applications, the likelihood function is unknown, but it is possible to obtain (approximate) samples from p(y|x) through a simulator. Such a problem is referred to as likelihood-free inference [3] or a black-box optimization problem [1]. If the problem is about finding the posterior distribution over *x* while only a simulator is available, then it is considered as an Approximate Bayesian Computation (ABC) problem, meaning that $p(y_{data}|x)$ is assumed to be given represented as the simulator.

## 3. Population-Based MCMC

Typically, a likelihood-free inference problem or an ABC problem is solved through sampling. One of the most well-known sampling methods is the Metropolis–Hastings algorithm [21], where the samples are generated from an ergodic Markov chain, and the target density is estimated via Monte Carlo sampling. In order to speed up the computations, it is proposed to run multiple chains in parallel rather than sampling from a single chain. This approach is known as population-based MCMC methods [22]. A population-based MCMC method operates over a joint state space with the following distribution:(3)$$p\left(x_{1},\ldots ,x_{C}\right)=\prod_{c\in C}p_{c}\left(x_{c}\right)$$
where C denotes the population of chains and at least one of $p_{c}\left(x_{c}\right)$ is equivalent to the original distribution we want to sample from (e.g., the posterior distribution $p(x|y_{data})$).

Given a population of chains, a question of interest is what is the best proposal distribution for an efficient sampling convergence. One approach is parallel tempering. It introduces an additional temperature parameter and initializes each chain at a different temperature [23,24]. However, the performance of the algorithm highly depends on an appropriate cooling schedule rather than a smart interaction between the chains. A different approach proposed by [25] relies on a suitable proposal that is able to adapt the shape of the population at a single temperature. We further expand on this idea by formulating population-based proposal distributions that are inspired by evolutionary algorithms.

### 3.1. Continuous Case

Reference [26] successfully formulated a new proposal called Differential Evolution Markov Chain (DE-MC) that combines the ideas of differential evolution and population-based MCMC. In particular, he redefined the DE-1 equation [16] by adding noise, ε, to it:(4)$$x_{new}=x_{i}+\gamma \left(x_{j}-x_{k}\right)+\varepsilon ,$$
where ε is sampled from a Gaussian distribution, $\gamma \in R_{+}$. The created proposal automatically implies the invariance of the underlying distribution, as the reversibility condition is satisfied:Reversibility is met, because the suggested proposal could be inverted to obtain $x_{i}$.

Furthermore, the created Markov chain is ergodic, as the following two conditions are met:Aperiodicity is met, because the Markov chain follows a random walk.Irreducibility is solved by applying the noise.

Hence, the resulting Markov chain has a unique stationary distribution. The results presented by [26] indicate an advantage of DE-MC over conventional MCMC with respect to the speed of calculations, convergence, and applicability to multimodal distributions, therefore positioning DE as an optimal method for choosing an appropriate scale and orientation of the jumping distribution for a population-based MCMC.

### 3.2. Discrete Case

In this paper, we focus on binary variables, because categorical variables could always be transformed to a binary representation. Hence, the most straightforward proposal for binary variables is the independent sampler that utilizes the product of Bernoulli:(5)$$q\left(x\right)=\prod_{d}B\left(\theta_{d}\right),$$
where $B(\theta_{d})$ denotes the Bernoulli distribution with a parameter $\theta_{d}$. However, the above proposal does not utilize the information available across the population; hence, the performance could be improved by allowing the chains to interact. Exactly this possibility we investigate in the following section.

## 4. Our Approach

### 4.1. Markov Kernels

We propose to utilize the ideas outlined by [26], but in a discrete space. For this purpose, we need to relate the DE-1 equation to logical operators, as now the vector *x* is represented by a string of bits, $X={\left\{0,1\right\}}^{D}$, and properly defined noise. Following [19], we propose to use the *xor*operator between two bits $b_{1}$ and $b_{2}$:(6)$$b_{1}⊗b_{2}=\left\{\begin{array}{cc} 1, & b_{1}\neq b_{2}\\ 0, & b_{1}=b_{2} \end{array}\right.$$
instead of the subtraction in (Equation 4). Next, we define a difference between two chains $x_{i}$ and $x_{j}$ as $\delta_{k}=x_{i}⊗x_{j}$ and a set of all possible differences between two chains, $\Delta =\{\delta_{k}:\forall_{x_{i},x_{j}\in C}\;\delta_{k}=x_{i}⊗x_{j}\}$ (a similar construction could be done for the continuous case). We can construct a distribution over $\delta_{k}$ as a uniform distribution:(7)$$q\left(\delta |C\right)=\frac{1}{\left|\Delta \right|}\sum_{\delta_{k}\in \Delta }I[\delta_{k}=\delta ],$$
where |Δ| denotes the cardinality of Δ and I[·] is an indicator function such that $I[\delta_{k}=\delta ]=1$ if $\delta_{k}=\delta$ and zero otherwise. Now, we can formulate a binary equivalence of the DE-1 equation by adding a difference drawn from q(δ|C):(8)$$x_{new}=x_{i}⊗\delta_{k}.$$However, the proposal defined in (Equation 8) is not a valid ergodic Markov kernel, as is shown in the following Proposition.

**Remark** **1.**
*The proposal defined in (Equation 8) fulfills reversibility and aperiodicity, but it does not meet the irreducibility requirement.*


**Proof.** Reversibility is met, as $x_{i}$ can be re-obtained by applying the difference to the left side of (Equation 8). Aperiodicity is met because the general setup of the Markov chain is kept unchanged (it resembles a random walk). However, the operation in (Equation 8) is deterministic; thus, it violates the irreducibility assumption.    □

The missing property of (Equation 8) could be fixed by including the following mutation (*mut*) operation:(9)$$x_{l}=\left\{\begin{array}{cc} 1-x_{l} & \mathrm{if}\;p_{flip}\geq u\\ x_{l} & \mathrm{otherwise} \end{array}\right.$$
where $p_{flip}\in \left(0,1\right)$ corresponds to an independent probability of flipping a bit and U(0,1) denotes the uniform distribution. Then, the following proposal could be formulated [19] as in Proposition 1.

**Proposition** **1.**
*The proposal defined as a mixture $q_{mut+xor}\left(x|C\right)=\pi q_{mut}\left(x|C\right)+\left(1-\pi \right)q_{xor}$(x|C), where π∈(0,1), $q_{mut}\left(x|C\right)$ is defined by (Equation 9) and $q_{xor}\left(x|C\right)$ is defined by (Equation 8), is a proper Markov kernel.*


**Proof.** Reversibility and aperiodicity were shown in Proposition 1. The irreducibility is met, because the *mut* proposal assures that there is a positive transition probability across the entire search space.    □

However, we notice that there are two potential issues with the mixture proposal *mut+xor*. First, it introduces another hyperparameter, π, that needs to be determined. Second, improperly chosen π could negatively affect the convergence speed, i.e., a fixed value that is either too frequent or scarce would drastically halt the convergence.

In order to overcome these issues, we propose to apply the *mut* operation in (Equation 9) directly to $\delta_{k}$, in a similar manner as the Gaussian noise is added to $\gamma (x_{i}-x_{j})$ in the proposition of [26]. As a result, we obtain the following proposal:(10)$$x_{new}=x_{i}⊗\left(\mathrm{mut}\left(\delta_{k}\right)\right).$$Importantly, this proposal fulfills all requirements for an ergodic Markov kernel.

**Proposition** **2.**
*The proposal defined in (Equation 10) is a valid ergodic Markov kernel.*


**Proof.** Reversibility and aperiodicity are met in the same manner as shown in Proposition 1. Adding the mutation operation directly to $\delta_{k}$ allows obtaining all possible states in the discrete space; thus, the irreducibility requirement is met.    □

We refer to this new Markov kernel for discrete random variables as the discrete differential evolution Markov chain (*dde-mc*).

### 4.2. Population-MCMC-ABC

Since we formulated a proposal distribution that utilizes a population of chains, we propose to use a population-based MCMC algorithm for the discrete ABC problems. The core of the MCMC-ABC algorithm is to use a proxy of the likelihood-function defined as an ϵ-ball from the observed data, i.e., $∥y-y_{data}∥\leq \epsilon$, where ϵ>0 and ∥·∥ is a chosen metric. The convergence speed and the acceptance rate highly depend on the value of ϵ [27,28,29]. In this paper, we consider two approaches to determine the ϵ value: (i) by setting a fixed value and (ii) by sampling ϵ∼Exp(τ) [30]. See the Appendix A for details.

A single step of the population-MCMC-ABC algorithm is presented in Algorithm 1. Notice that in Line 5, we take advantage of the symmetricity of all the proposal. Moreover, in the procedure, we skip an outer loop over all chains for clarity. Without loss of generality, we assume a simulator to be a probabilistic program denoted by $\tilde{p}\left(y|x\right)$.
**Algorithm 1** Population-MCMC-ABC.1:Given $x\in {\left\{0,1\right\}}^{D}$2:$x^{\prime }∼q\left(x|C\right)$             ▹ Either (Equation 5), *mut+xor* or *dde-mc*.3:Simulate $y∼\tilde{p}\left(y|x^{\prime }\right)$.4:**if**$∥y-y_{data}∥\leq \epsilon$**then**5:    $\alpha =\min\{1,\frac{p(x^{\prime })}{p(x)}\}$6:    u∼U(0,1)7:    **if**
u≤α
**then**8:        $x=x^{\prime }$9:**return***x*

## 5. Experiments

In order to verify our proposed approach, we use four test-beds:QMR-DT network (likelihood-based case): First, we validate the novel proposal, *dde-mc*, on a problem when the likelihood is known.QMR-DT network (likelihood-free case): Second, we verify the performance of the presented proposal by modifying the first test-bed as a likelihood-free problem.Binarized Neural Network Learning: Third, we investigate the performance of the proposed approach on a high-dimensional problem, namely learning binary neural networks.Neural architecture search: Lastly, we consider a problem of Neural Network Architecture Search (NAS).

With each test-bed, we increase the complexity of the problem. Hence, the number of iterations chosen varies per experiment. The code of the methods and all experiments is available at the following link: https://github.com/IlzeAmandaA/ABCdiscrete (accessed on 5 March 2021).

### 5.1. A Likelihood-Based QMR-DT Network

#### 5.1.1. Implementation Details

The overall setup was designed as described by [19], i.e., we considered a QMR-DT network model. The architecture of the network follows a two-level or bipartite graphical model, where the top level of the graph contains nodes for the diseases and the bottom level contains nodes for the findings [31]. The following density model captures the relations between the diseases (*x*) and findings (*y*):(11)$$p\left(y_{i}=1|x\right)=1-\left(1-q_{i0}\right)\prod_{l}{\left(1-q_{il}\right)}^{x_{l}}$$
where $y_{i}$ is an individual bit of string *y* and $q_{i0}$ is the corresponding leak probability, i.e., the probability that the finding is caused by means other than the diseases included in the QMR-DT model [31]. $q_{il}$ is the association probability between disease *l* and finding *i*, i.e., the probability that the disease *l* alone could cause the finding *i* to have a positive outcome. For a complete inference, the prior p(x) is specified. We follow the assumption made by [19] that the diseases are independent:(12)$$p\left(x\right)=\prod_{l}p_{l}^{x_{l}}{\left(1-p_{l}\right)}^{\left(1-x_{l}\right)}$$
where $p_{l}$ is the prior probability for disease *l*.

We compare the performance of the *dde-mc* kernel to the *mut* proposal, the *mut-xor* proposal, the *mut+crx* proposal (see [19] for details), and the independent sampler (*ind-samp*) as in (Equation 5) with sampling probability $\theta_{d}=0.5$. We expect the DE-inspired proposals to outperform *ind-samp*, and *dde-mc* to perform similarly, if not surpass, *mut+xor*. Out of the possible parameter settings we investigate, the following population sizes $C=\left\{8,12,24,40,60\right\}$, as well as bit-flipping probabilities $p_{flip}$ = $\left\{0.1,0.05,0.01,0.005\right\}$. All experiments were run for 10,000 iterations, as in earlier work by [19], it was observed that the performance differences after 10,000 steps were negligible, and initial experiments revealed that in the current work, all proposals approximately converged at this mark. Furthermore, the performance was validated over 80 random problem instances, and the resulting mean and its standard error are reported.

In this experiment, we used the error that is defined as the average Hamming distance between the real values of *x* and the most probable values found by the population-MCMC with different proposals. The number of diseases was set to m=20, and the number of findings was n=80.

#### 5.1.2. Results and Discussion

DE-inspired proposals, *dde-mc* and *mut+xor*, are superior to kernels stemming from genetic algorithms or random search, i.e., *mut+crx*, *mut*, and *ind-samp* (Figure 1). In particular, *dde-mc* converged the fastest (see the first 4000 evaluations in Figure 1), suggesting that an update via a single operator rather than a mixture is most effective. As expected, *ind-samp* requires many evaluations to obtain a reasonable performance. Even more so, the obtained difference in wall-clock time between *dde-mc* and *ind-samp* was negligible, 148 versus 117 min, respectively, even though the computational complexity of the new method is theoretically higher: given a search space of ${\left\{0,1\right\}}^{D}$, the *dde-mc* proposal costs O(D), while the time complexity of *ind-samp* is O(1).

Based on the obtained results, the subsequent experiments were carried out only with *dde-mc*, *mut+xor*, and *ind-samp* as a baseline. *mut+crx* and *mut* were not selected due to to their very slow convergence with high-dimensional problems.

### 5.2. A Likelihood-Free QMR-DT Network

#### 5.2.1. Implementation Details

In this test-bed, the QMR-DT network is redefined as a simulator model, i.e., the likelihood is assumed to be intractable. The Hamming distance is selected as the distance metric, but due to its equivocal nature for high-dimensional data, the dimensionality of the problem is reduced. In particular, the number of diseases and observations (i.e., findings) are decreased to 10 and 20, respectively, while the probabilities of the network are sampled from a beta distribution, Beta(0.15,0.15). The resulting network is more deterministic as the underlying density distributions are more peaked; thus, the stochasticity of the simulator is reduced. Multiple tolerance values are investigated to find the optimal settings, $\epsilon =\left\{0.5,0.8,1.,1.2,1.5,2.\right\}$, respectively. The minimal value is chosen to be 0.5 due to variability across the observed data $y_{data}$. Additionally, we checked sampling ϵ from the exponential distribution. All experiments were cross-evaluated 80 times, and each experiment was initialized with different underlying parameter settings.

#### 5.2.2. Results and Discussion

First, for the fixed value of ϵ, we notice that *dde-mc* converged faster and to a better (local) optimum than *mut+xor*. However, this effect could be explained by a lower dimensionality of the problem compared to the first experiment. Second, utilizing the exponential distribution had a profound positive effect on the convergence rate of both *dde-mc* and *mut+xor* (Figure 2). This confirmed the expectation that an adjustable ϵ has a better balance between exploration and exploitation. In particular, ϵ∼Exp(2) brought the best results with *dde-mc* converging the fastest, followed by *mut+xor* and *ind-samp*. This is in line with the corresponding acceptance rates for the first 10,000 iterations (Table 1), i.e., the use of a smarter proposal allows increasing the acceptance probability, as the search space is investigated more efficiently.

Furthermore, the final error obtained by the likelihood-free inference approach is comparable with the results reported for the likelihood-based approach (Figure 1 and Figure 2). This is a positive outcome as any approximation of the likelihood will always be inferior to an exact solution. In particular, the final error obtained by the *dde-mc* proposal is lower; however, this is accounted for by the reduced dimensionality of the problem. Interestingly, despite approximating the likelihood, the computational time only increased twice, while the best performing chain was already identified after 4000 evaluations (Figure 3).

Lastly, the obtained results were validated by comparing the true approximate posterior distribution to the approximate posterior distribution of the last five generations of the multi-chain ensemble. In Figure 4, the negative logarithm of the posterior distribution is plotted. The main conclusion is that all proposals converge towards the approximate posterior, yet the obtained distributions are more dispersed.

### 5.3. Binary Neural Networks

#### 5.3.1. Implementation Details

In the following experiment, we aimed at evaluating our approach on a high-dimensional optimization problem. We trained a Binary Neural Network (BinNN) with a single fully-connected hidden layer on the image dataset of ten handwritten digits (MNIST [32]). We used 20 hidden units, and the image was resized from 28px × 28px to 14px × 14px. Furthermore, the image was converted to polar values of +1 or −1, while the network was created in accordance to [33], where the weights and activations of the network were binary, meaning that they were constrained to +1 or −1 as well. We simplified the problem to a binary classification by only selecting two digits from the dataset. As a result, the total number of weights equaled 3940. We used the tanh activation function for the hidden units and the sigmoid activation function for the outputs. Consequently, the distance metric becomes the classification error:(13)$$∥y_{data}-y∥=1-\frac{1}{N}\sum_{n=1}^{N}I\left[y_{n}=y_{n}\left(x\right)\right],$$
where *N* denotes the number of images, I[·] is an indicator function, $y_{n}$ is the true label for the *n*-th image, and $y_{n}\left(x\right)$ is the *n*-th label predicted by the binary neural net with weights *x*.

For the Metropolis acceptance rule, we define a Boltzmann distribution over the prior distribution of the weights *x* inspired by the work of [34]:(14)$$p\left(x\right)=\frac{h(x)}{\sum_{i}h\left(x_{i}\right)},$$
where $h\left(x\right)=exp\left(-\frac{1}{D}\sum_{i=1}^{D}x_{i}\right)$ and *D* denotes the dimensionality of *x*. As a result, the prior distribution acts as a regularization term as it favors parameter settings with fewer active weights. The distribution is independent of the data *y* thus, the partition function $\sum_{i}h\left(x_{i}\right)$ cancels out in the computation of the Metropolis ratio:(15)$$\alpha =\frac{p(x^{\prime })}{p(x)}=\frac{h(x^{\prime })}{h(x)}.$$

The original dataset consists of 60,000 training examples and 10,000 test examples. For our experiment, we selected the digits 0 and 1; hence, the dataset size was reduced to 12,665 training and 2115 test examples. Different tolerance values were investigated to obtain the best convergence, ranging from 0.03 to 0.2, and each experiment was run for at least 200,000 iterations. All experiments were cross-evaluated five times. Lastly, we evaluated the performance by computing both the minimum test error obtained by the final population, as well as the test error obtained by using a Bayesian approach, i.e., we computed the true predictive distribution via majority voting by utilizing an ensemble of models. In particular, we selected the five last updated populations, resulting in 5 × 24 × 5 = 600 models per run, and we repeated this with different seeds 10 times.

Because the classification error function in (Equation 13) is non-differentiable, the problem could be treated as a black-box objective. However, we want to emphasize that we do not propose our method as an alternative to gradient-based learning methods. In principle, any gradient-based approach will be superior to a derivative-free method, as what a derivative-free method tries to achieve is to implicitly approximate the gradient [1]. Therefore, the purpose of the presented experiment is not to showcase a state-of-the-art classification accuracy, as that already has been done with gradient-based approaches for BinNN [33], but rather showcase the population-MCMC-ABC applicability to a high-dimensional optimization problem.

#### 5.3.2. Results and Discussion

For the high-dimensional data problem, the *mut+xor* proposal converged the fastest towards the optimal solution in the search space (Figure 5). In particular, the minimum error on the training set was already found after 100,000 iterations, and a tolerance threshold of 0.05 had the best trade-off between the Markov chain error and the likelihood approximation bias.

With respect to the error within the entire population (Figure 6), *dde-mc* converged the fastest, although its performance was on par with *ind-samp*. In general, the drop in performance with respect to the convergence rate of the entire population could be explained by the high dimensionality of the problem, i.e., the higher the dimensionality, the more time is needed for every chain to explore the search space. This observation was confirmed by computing the test error via utilizing all the population members in a majority-voting setting. In particular, the test error based on the ensemble approach was alike across all three proposals, yet the minimum error (i.e., for a single best model) was better for *dde-mc* and *mut+xor* compared to *ind-samp* (Table 2). This result suggests that there seems to be an added advantage of utilizing DE-inspired proposals in faster convergence towards a local optimal solution.

### 5.4. Neural Architecture Search

#### 5.4.1. Implementation Details

In the last experiment, we aimed at investigating whether the proposed approach is applicable for efficient neural architecture search. In particular, we made use of the NAS-Bench-101 dataset, the first public architecture dataset for NAS research [20]. The dataset is represented as a table, which maps neural architectures to their training and evaluations metrics, and as such, it represents an efficient solution for querying different neural topologies. Each topology is captured by a directed acyclic graph represented by an adjacency matrix. The number of vertices was set to seven, while the maximum amount of edges was nine. Apart from these restrictions, we limited the search space by constricting the possible operations for each vertex. Consequently, the simulator was captured by querying the dataset, while the distance metric now was simply the validation error. The prior distribution was kept the same as for the previous experiment.

Every experiment was run for at least 120,000 iterations, with five cross-evaluations. To find the optimal performance, the following tolerance threshold values were investigated $\epsilon =\left\{0.01,0.1,0.2,0.3\right\}$. As we are approaching the problem as an optimization task, the aim is to find a chain with the lowest test error, rather than covering the entire distribution. Therefore, to evaluate the performance, we plot the minimum error obtained through the training process, as well as the lowest test error obtained by the final population.

#### 5.4.2. Results and Discussion

*dde-mc* identified the best solution the fastest with ϵ set to ϵ∼Exp(0.2) (Figure 7). The corresponding test error is reported in Table 3, and it follows the same pattern, namely *dde-mc* is superior. Interestingly, here, the *mut+xor* proposal performed almost on par with the *ind-samp* proposal for the first 10,000 iterations, and then, both methods converged to almost the same result. Our proposed Markov kernel obtained again the best result, and also it was the fastest.

## 6. Conclusions

In this paper, we note that there is a gap in the available methods for likelihood-free inference on discrete problems. We propose to utilize ideas known from evolutionary computing similarly to [26], in order to formulate a new Markov kernel, *dde-mc*, for a population-based MCMC-ABC algorithm. The obtained results suggest that the newly designed proposal is a promising and effective solution for intractable problems in a discrete space.

Furthermore, Markov kernels based on differential evolution are also effective to traverse a discrete search space. Nonetheless, great attention has to be paid to the choice of the tolerance threshold for the MCMC-ABC methods. In other words, if the tolerance is set too high, then the performance of the DE-based proposals drops to that of an independent sampler, i.e., the error of the Markov chain is high. For high-dimensional problems, the proposed kernel seems to be most promising; however, its population error becomes similar to that of *ind-samp*. This is accounted for by the fact that for high dimensions, it takes more time for the entire population to converge.

In conclusion, we would like to highlight that the present work offers new research directions:Alternative ABC algorithms like SMC should be further investigated.In this work, we focused on calculating distances in the data space. However, utilizing summary statistics is almost an obvious direction for future work.As the whole algorithm is based on logical operators and the input variables are also binary, the algorithm could be encoded using only bits, thus saving considerable amounts of memory storage. Consequently, any matrix multiplication could be replaced by an XNORoperation followed by a sum, thus reducing the computation costs and possibly allowing implementing the algorithm on relatively simple devices. Therefore, a natural consequence of this work would be a direct hardware implementation of the proposed methods.In this paper, we outline a number of potential applications of the presented methodology and indicate that the obtained results are of great practical potential. From the optimization perspective, a discrete ABC gives an opportunity to solve a problem in a principled manner. This is extremely important for applications associated with deep learning, e.g., NAS [20,35], neural network quantization, and learning binary neural networks, but also in other domains like topology or relationship discovery in biological networks (e.g., Boolean networks) [36]. Moreover, ABC as a Bayesian framework allows calculating model evidence that is crucial for model selection. In practice, very often, a problem is of combinatorial (discrete) nature, e.g., contamination control or pest control [35]. Therefore, our approach could be seemingly applied without the necessity of dequantizing a problem.

## Figures and Tables

**Figure 1 entropy-23-00312-f001:**
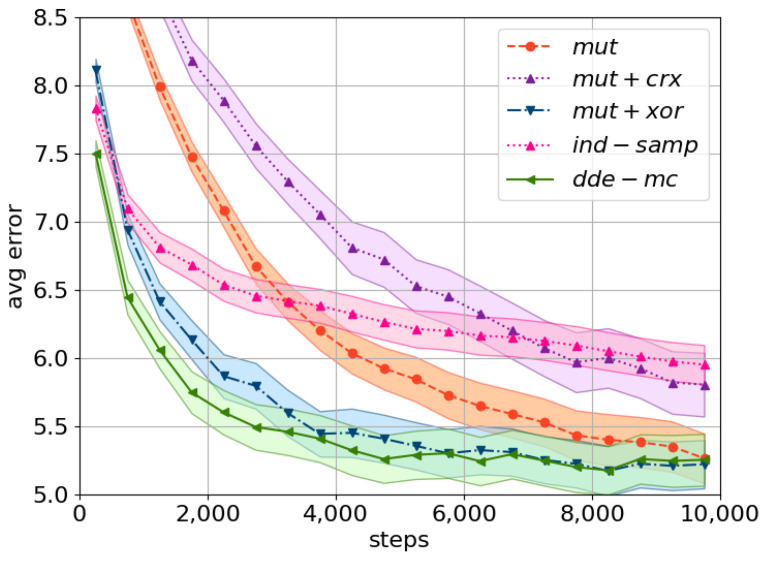
A comparison of the considered proposals using the population average error. The obtained mean and its corresponding standard error (shaded area) across 80 random problem instances are plotted. The following settings were used: C=24, $p_{flip}=0.01$, $p_{cross}=0.5$. The corresponding equations for each proposal are as follows: *mut* as in (Equation 9), *ind-samp* as in (Equation 5), *dde-mc* as in (Equation 10), and *mut+xor*, *mut+crx* as in [19].

**Figure 2 entropy-23-00312-f002:**
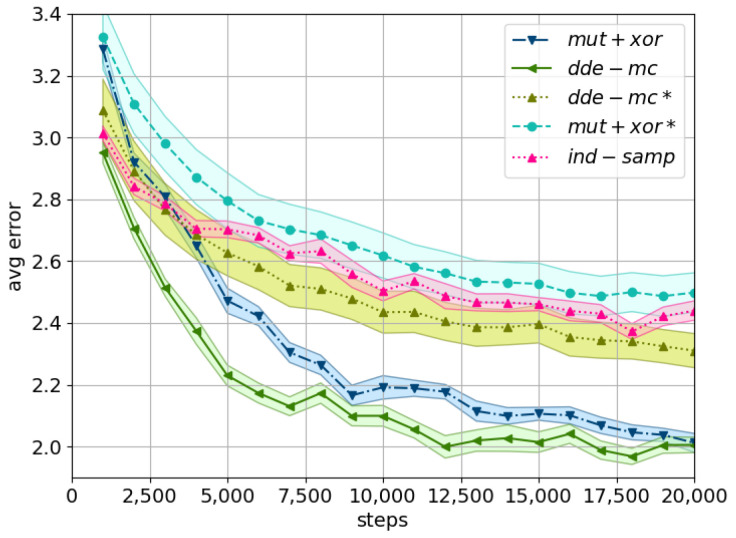
A comparison of the considered proposals using the population error for exponentially adjusted ϵ and the fixed ϵ (indicated by *). The shaded area corresponds to the standard error across 80 random problem instances. The parameter settings are as follows: C=24, $p_{flip}=0.01$, ϵ=2.0. The following equations describe the proposal distributions utilized in Algorithm 1: *ind-samp* as in (Equation 5), *dde-mc* as in (Equation 10), and *mut+xor* as in [19].

**Figure 3 entropy-23-00312-f003:**
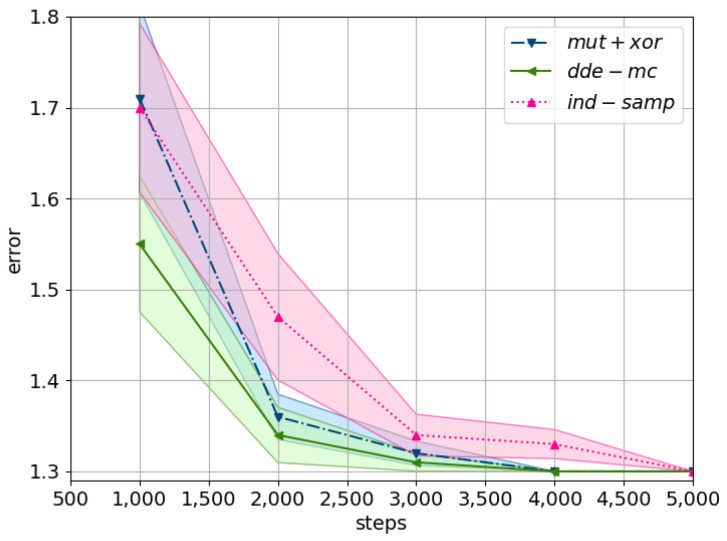
A comparison of the considered proposal using the minimum average error (i.e., the lowest error found by the population) on QMR-DTwith adjusted ϵ. The shaded area corresponds to the standard error across 80 random problem instances. The parameter settings are as follows: C=24, $p_{flip}=0.01$, ϵ=2.0. The corresponding equations represent the proposal distributions: *ind-samp* in (Equation 5), *dde-mc* in (Equation 10), and *mut+xor* as in [19].

**Figure 4 entropy-23-00312-f004:**
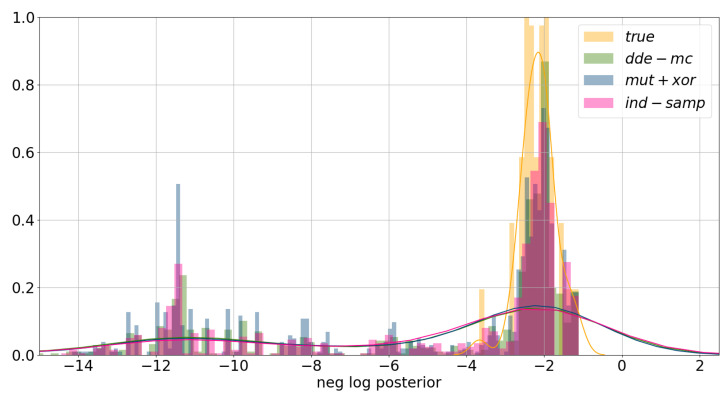
Approximate posterior distribution. The approximate posterior distribution, $p\left(x|y_{data}\right)\approx p\left(y_{data}|x\right)*p\left(x\right)$, was computed using the last population of each chain for all 80 random problem instances. To reconstruct the true posterior, the true underlying parameters were used.

**Figure 5 entropy-23-00312-f005:**
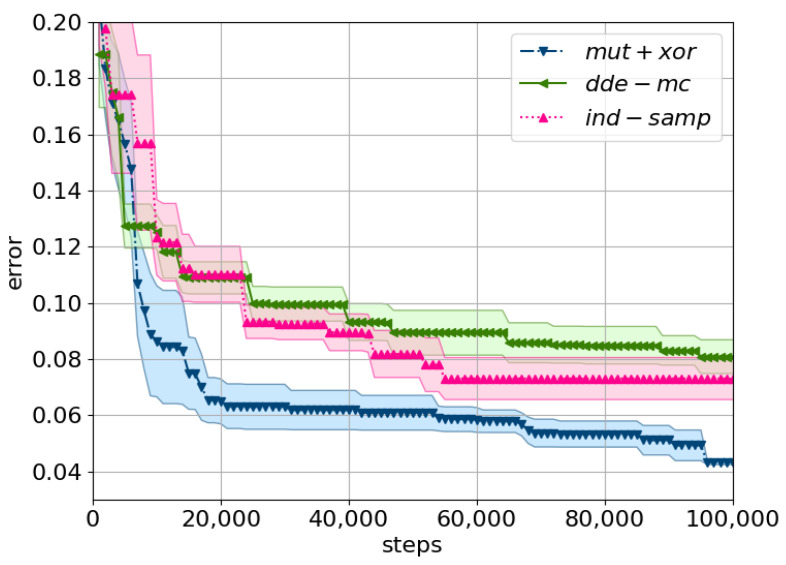
A comparison of the considered proposals using the minimum training error on MNIST. The mean minimum error across five cross-evaluations is plotted with the shaded area corresponding to the standard error. Tolerance is set to ϵ=Exp(0.05), with the prior and the Metropolis ratio as described in (Equation 14) and (Equation 15). The following equations describe the proposals: *ind-samp* in (Equation 5), *dde-mc* in (Equation 10), and *mut+xor* as in [19].

**Figure 6 entropy-23-00312-f006:**
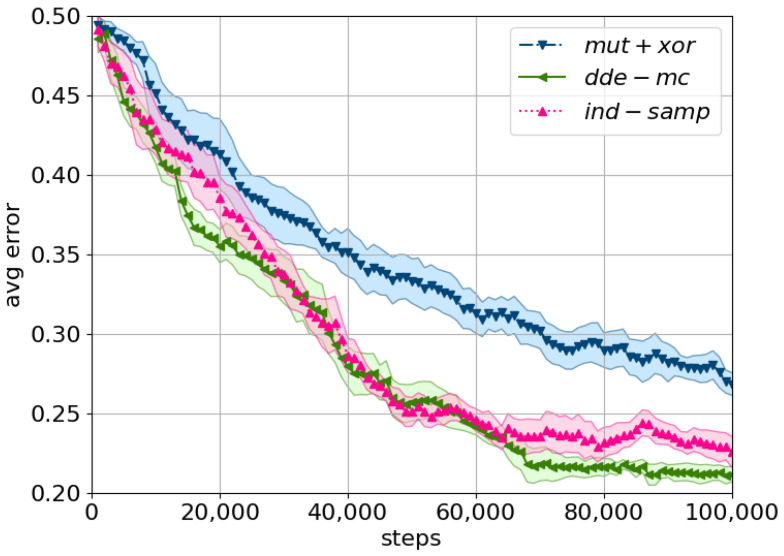
A comparison of the considered proposals using the avg. training error on MNIST. The mean population error across five cross-evaluations is plotted with the shaded area corresponding to the standard error. Tolerance is set to ϵ=Exp(0.05), with the prior and the Metropolis ratio as described in (Equation 14) and (Equation 15). The following equations describe the proposals: *ind-samp* in (Equation 5), *dde-mc* in (Equation 10), and *mut+xor* as in [19].

**Figure 7 entropy-23-00312-f007:**
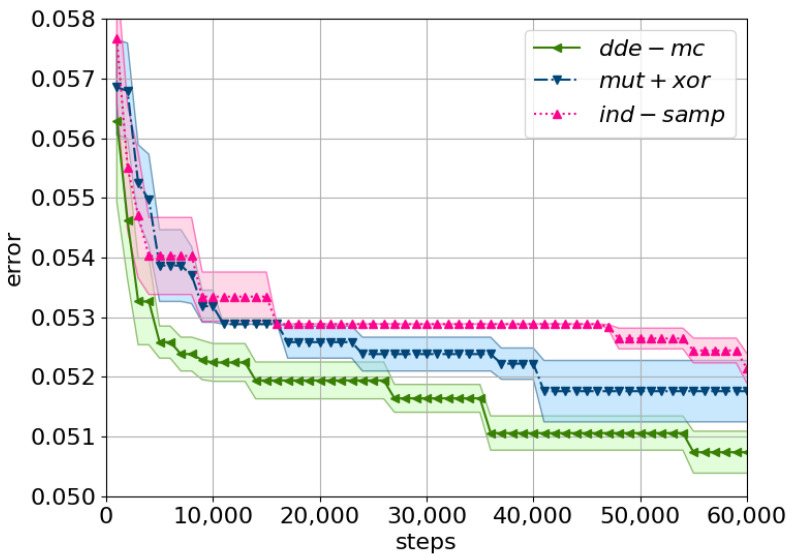
A comparison of the considered proposals using the minimum training error on NAS-Bench-101. The mean minimum error with its corresponding standard error (shaded area) across five cross-evaluations is plotted. Tolerance is set to ϵ=Exp(0.2). The prior distribution is as described in (Equation 14), with the corresponding Metropolis ratio (Equation 15). The following equations describe the proposals: *ind-samp* in (Equation 5), *dde-mc* in (Equation 10), and *mut+xor* as in [19].

**Table 1 entropy-23-00312-t001:** Percentage of acceptance ratio, α.

Proposal	Mean (std)
*dde-mc*	24.47 (1.66)
*mut+xor*	25.81 (1.38)
*ind-samp*	13.14 (0.33)

**Table 2 entropy-23-00312-t002:** Test error of BinNN on MNIST.

Proposal	Error (ste)	
	*Single Best*	*Ensemble*
*dde-mc*	0.045 (0.002)	0.013 (0.001)
*mut+xor*	0.046 (0.002)	0.014 (0.002)
*ind-samp*	0.051 (0.002)	0.012 (0.001)

**Table 3 entropy-23-00312-t003:** Test error on NAS-Bench-101.

Proposal	Error (ste)
*dde-mc*	0.058 (0.001)
*mut+xor*	0.060 (<0.001)
*ind-samp*	0.062 (<0.001)

## Data Availability

The code used in this paper is available at: https://github.com/IlzeAmandaA/ABCdiscrete (accessed on 5 March 2021).

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
