# Peer review of "Approximate Bayesian Computation for Discrete Spaces"

_entropy, 2021, doi:10.3390/e23030312_

Round 1
Reviewer 1 Report
In the present paper, the authors propose an adjusted population-based MCMC ABC method by re-defining the standard ABC parameters to discrete ones and by introducing a novel Markov kernel. The presented solution is inspired by differential evolution and the proposed approach is illustrated for dealing with binary problems. According to their results, the high potential of the proposed framework is confirmed, while the superiority of the new Markov kernel has been also revealed. The present research work is well-written and constitutes a significant contribution on the topic. Therefore, I recommend the publication of the paper after addressing appropriately the following issues.
- Define all symbols appeared in Equation (7).
- Line 100: The word “Hoever” should corrected as “However”.
- Proposition 1 could be presented as a Remark or Lemma.
- The authors claim that “All experiments are run for 10,000 iterations”. The authors are recommended to give some evidence about the accuracy of their numerical results and/or provide relative references where a similar number of iterations have been used.
- Below Equation (14), the symbol D should be defined.
- Throughout the lines of Section 5, authors run their experiments based on different number of iterations each time. The authors are encouraged to explain their different choices.
Reviewer 2 Report
The manuscript presents approximate Bayesian computations for discrete spaces comparing a number of available and new algorithms and approaches. Although the subject is clearly hot and important and i do support publication of this interesting material, the manuscript still needs a moderate revision prior to publication.I would advice the authors to list all the abbreviations in the end of the manuscript: as they are plenty, such as table or list would simplify the life of the reader a lot. With regard to Bayesian model-assessment algorithms for a set of mathematical models of diffusion of a stochastic variable, based on a given input set of the time series, i would like to refer the authors to the thorough recent methodological and real-data-analysis studies [ https://doi.org/10.1039/C8CP04043E ] and [https://doi.org/10.1039/C8SM02096E ]. The approaches developed in these two references enabled definite model-prediction and parameter-estimation statements regarding the realizable models of both normal and anomalous diffusion. These two very relevant recent studies can be described at certain length in the revised version of the current manuscript. The graphs of this manuscript are not easy to read. Please remove that gray background, keep the same notations for the symbols denoting the respective algorithms/conditions, and please provide the detailed captions, including the values of all the parameters and the equation numbers of all formulae plotted. The graphical information should give the reader the instant picture of what is done in the paper, without spending lots of time reading the entire text. Lastly, the authors can discuss in more details what are real physical systems they have in mind to apply these algorithms and, in case same real data were already analyzed and certain assessment of performance was already conducted, this can be explicitly included in the revised version. This way the future paper (which is very methodological not) will also become of interest for the experimental community (of, for instance, single-particle tracking people).
